# Issues of EU Member Nations' Shared Sovereignty, Institutions, and Economic Development

## Ismatilla Mardanov

Department of Management, Southeast Missouri State University, Cape Girardeau, MO 63701, USA; imardanov@semo.edu

**Abstract:** To investigate the effects of the European Union's (EU) member nations' shared sovereignty on economic growth. The member nations have lost substantial political and economic independence (sovereignty) and democracy. Therefore, their governments cannot facilitate rapid economic growth in their countries, affecting the EU as a whole. Data from the World Bank, institutional research entities, and the EU were utilized. The dependent variable is economic growth, and the independent and moderating variables are mainly institutions and the European Sovereignty Index. Shared sovereignty and its specific categories and foreign direct investment (FDI) outflows negatively impact economic development in the EU. Shared sovereignty negatively moderates the relationship between political rights and economic development and between FDI outflows and economic development. Democracy in member nations is formal rather than real. The present study focused on the EU's problems rather than its achievements and empirically investigated the direct and moderating effects of national sovereignty and member-country institutions on member-country economic growth. This focus and the nature of the investigation constitute the originality of the present study and reduce the gap in the literature about the effects of sovereignty, institutions, and capital spillovers (FDI outflows) on economic growth in Europe. The value of the study is in its findings, which should trigger holistic research efforts on the pros and cons of the EU for Europe, democracy, the economy, and the world.

**Keywords:** the European Union; shared sovereignty; democracy; FDI; economic growth





## 1. Introduction

The concept of a United States of Europe is not a recent one, as it has been mentioned by prominent figures throughout history such as Napoleon Bonaparte, Victor Hugo, Giuseppe Garibaldi, and John Stewart Mill. Winston Churchill expressed his concerns about Europe in 1946 and suggested the formation of a Council of Europe as the first step towards creating a regional structure known as the United States of Europe (USE).

There is substantial research on the positive outcomes of EU membership, including the rise of Europe and the advantages of the European Union (Dzemydaite 2021; Pastor et al. 2018; Greenaway et al. 2000; Lock 2009; Daniel and Shiamptanis 2008; Acemoglu et al. 2005). The European Union has had a positive impact on new member nations, with per capita GDP increasing in all of them (Baneliene 2013) and more significantly in some, such as Slovenia. However, some older member countries, such as Greece, had negative GDP growth rates about a decade after the last global financial crisis. At the same time, non-member European countries, such as Switzerland and Norway, achieved significantly higher growth rates. This was due to the EU's economic specialization policies (Dzemydaite 2021), whereby Greece's specialization was in agriculture and tourism. Over the past ten years, pre-pandemic, Europe's economic growth rate was between one and two percent, which was comparable to the former Soviet Union's economic growth rates set by their centralized planning system.

The present study aims to analyze the issues and problems of the European Union and explore methods to address them. Therefore, it does not focus on the positive aspects of the EU, which is a unique structural phenomenon.

The European Union is a unique political entity that is neither a confederation nor a federation but is moving toward becoming a federation (Hazak 2012). Its governing system is based on the Lisbon Treaty, and its structural formation can be considered experimental. The European Parliament's representation is based on population size, with larger countries having more representatives and more influence on the legislative process and outcomes. Meanwhile, the European Council is composed of country executive branch leaders. However, unlike the American Senate, the EU has no upper legislative chamber with equal representation of member states, resulting in smaller countries having less influence despite delegating a significant part of their sovereignty to the EU. This discrepancy is undermining member-state democracies and diminishing their rights.

Germany has advocated for a single constitution for the EU (Thym 2009), which could create a more legitimate union with a more effective and efficient government. However, the idea of referenda was blocked by voters in France and the Netherlands. The EU's institutional setup and legitimacy among its citizens have been questioned in the literature (Dimitrova 2010), with some attributing the problems to national polities (Schmidt 2006). The EU's super governance, inappropriately shared sovereignty, national polities, and cultural differences could be factors contributing to the low economic growth rates of the EU for many years. In the second decade of the Eurozone, several countries experienced significant financial difficulties (Kuforiji 2016). Additionally, the structural setup of some peripheral countries' economies under the division of labor in the EU was a primary cause of the deep economic crises in those countries.

The gap between research and practice is evident in the EU, where research has been reluctant to analyze the issues and problems of this conglomerate that tightly controls member states' political, economic, and social systems. The objectives of this study are to analyze the pros and cons of member nations delegating their political and economic sovereignty and the effects of shared sovereignty (De Burca 2003) on economic development. Sovereignty can serve as an instrumental or moderating variable that strengthens or weakens the relationship between economic and political institutions and economic development. This study will also question the EU's feasibility and legitimacy (Cohen 2012), and the erosion of democracy in member nations.

In his article for the Harvard Business Review, Bill Lee (2013) said, "How long can this go on?" He continued: "According to a recent article in *The Wall Street Journal*, the 17-nation Eurozone remains the 'weakest link' in our global economy after years of economic stagnation. It is mired in high unemployment, plagued with stalled or contracting economies, and paralyzed by political dysfunction. Similarly, *The Economist* lambasts eerily complacent EU leaders for sleepwalking through an economic wasteland." The excessive delegation of political and economic sovereignty to the European Parliament and the EU's executive bodies has contributed significantly to member states' economic and social problems, according to Grosse (2016).

In free societies, people recognize that their country belongs to them and they elect politicians to develop their economy and improve their lives. However, this belongingness is under question in the member nations of the EU, where there is unequal representation in the European Parliament from member nations and the EU's executive bodies.

## 2. The Theoretical Framework

### 2.1. Shared Sovereignty and Economic Development

The European Economic Community (EEC) was established in 1957 with the aim of achieving economic integration through a single market and customs union, and to create a centralized foreign trade policy to withstand global competition. While forming alliances, partnerships, and communities among countries is a natural and justified step, joining such communities requires sharing national sovereignty with a common body that makes decisions for the people of member nations. As the community evolves into a union, such as the European Union, member nations lose an essential part of their sovereignty.

This is particularly true when they enter a common currency zone as they surrender the overwhelming share of their national sovereignty.

The present study supports the classical theory of state sovereignty based on West-phalian or Vattelian sovereignty, which is also the foundation of the Montevideo Convention (1933) on the Rights and Duties of States. External, Westphalian, or Vattelian sovereignty refers to a state's autonomy or self-determination regarding external players. On the other hand, internal or domestic sovereignty is defined by an institutional and constitutional system that ensures the effective exercise of power according to the sovereign's will (Krasner 1999).

The Montevideo Convention officially defined external sovereignty. According to Article 1, a state as a person of international law must possess the following qualifications: a permanent population, a defined territory, a government, and the capacity to enter into relations with other states. The internal aspect of sovereignty is defined in Article 3, which states that a state has the right to defend its integrity and independence, provide for its conservation and prosperity, organize itself as it sees fit, legislate upon its interests, administer its services, and define the jurisdiction and competence of its courts. The exercise of these rights is limited only by the exercise of the rights of other states according to international law. However, the convention does not envision the formation of unions such as the European Union.

Initially, sovereignty was associated with the indivisibility and inalienability of ultimate power (Hinsley 1986), which was the state. Francis Hinsley (1986, p. 223) stated that " . . . in modern times—the rise of legislatures, the introduction of representation, the extension of suffrages and the insertion of constitutional features into the basis, the composition and the procedures of government—necessitated the notion that sovereignty resided in the body-politic as a means of preserving the precondition of effective action in and for the community, the sovereignty of the state."

In other words, sovereignty exists for the government/state and the community so that it can address problems using the government as the ultimate instrument. Furthermore, Francis Hinsley says that the sovereign state must have coercive authority to make political and legal systems function. The member states of the EU have no real democracy: political processes are formal and inconsequential. A winning political party or block follows the directives of the European Commission because member states share their external and internal sovereignty with the European Union.

Neil MacCormick (1993) argues that since no nation-state in the EU is in "a position such that all the power exercised internally in it, whether politically or legally, derives from purely internal sources," none can be regarded any longer as a sovereign state. He asked "whether or not there are any Sovereign States here, now, anymore?" He said: "I am going to answer that negatively" (MacCormick 1993). Later he powerfully affirmed his stance on sovereignty in the European Union: "absolute or unitary sovereignty is absent from the legal and political setting of the European Community—neither politically nor legally is any member state in possession of ultimate power over its internal affairs" (MacCormick 1999). A European Court of Justice ruling in 1963 claimed that the arrangement "creates a new legal order . . . for the benefit of which the States have limited their sovereign rights, albeit within limited fields" (Judgment of the Court 1963). Loughin (2016) suggested that because the EU arrangement is permanent, and it is no longer within the political authority of a member state to exercise the procedure for withdrawal from treaty arrangements, no issue of ultimate authority—and no question of sovereignty—is involved.

The view of infringement of the UK's sovereignty due to the Maastricht Treaty of 1993 was predominant until 2018, when the UK decided to exit the EU. MacCormick (1993) wrote: "There is a widespread, but perhaps misguided, belief that there are a lot of sovereign states in the world, that this is a good thing, that the United Kingdom is one, and that it will be a bad thing if the UK ceases to be so. It is also a majority view that if the United Kingdom has a constitution at all, its central pillar is the principle of the sovereignty of Parliament. No sovereignty, no constitution; no constitution, no UK."

The primary view of the EU in the sovereignty research field is that European integration excessively limited the member states' Westphalian (or Vattelian) sovereignty, or even deprived them of it (Krasner 1999). Due to integration, the national sovereignty of member states has become a pooled resource or common sovereignty (Czaputowicz 2013, 2015). In this context, the paradox is that national elections are inconsequential formalities because the winning party and new government cannot address the nation's problems without the EU's approval.

*2.2. Institutions and Economic Development*

In the EU's Eurozone, no political party can promise voters economic development programs or projects. The European Central Bank rejects any national projects if it decides that those programs are not a priority for the EU. Additionally, the absence of national currency is a massive distortion of the economic sovereignty of member nations. The European Commission would blame the national governments for being selfish. The European Central Bank blames member country governments for making "wrong political decisions with the aim of re-election and maintaining the political power" (Tkáčová et al. 2018). Officials in the European Commission state that democratic outcomes (elections, referenda, etc.) must not be allowed to challenge the economic policy of the Eurozone, which creates tensions in the member countries (Markantonatou et al. 2018). Without economic sovereignty, political sovereignty does not exist. In all of the member nations, party-based democracy is in crisis (Bickerton et al. 2022). The European Sovereignty Index (Puglierin and Zerka 2022) confirms the limited institutional sovereignty of member nations.

While the economic institutions of EU member nations are rated high in the economic freedom index, overall economic integration and many specific centralized economic decisions did not contribute significantly to their prosperity. Per capita GDP in the first ten years has increased by only ten percent in member nations that entered the EU from 1973 to 2004; the effects of deep integration into the EU were positive but not overwhelming for some countries, and not so favorable for other member countries in terms of per capita GDP increase and labor productivity (Campos et al. 2019). Europe has a relatively stable financial system, and the literature confirms a long-term equilibrium relationship between banking development, stock market development, and economic development (Wu et al. 2009). However, that stability could have been more consistent over the decades, and the current institutional system of the EU may not help dynamic economic development as desired. The quality of institutions matters (Zeqiraj et al. 2022), and the EU has high-quality institutions. However, much progress remains to be made before they support true democracy, political freedom of member nation citizens, and economic competitiveness. Europe is losing trade and investment competition to China and the United States because of weakened national democratic institutions before the EU's governing bodies. The EU is implementing cohesion policies (Farole et al. 2011). The most recent policy includes the years 2021 to 2027. The EU Cohesion Policy strengthens the European Union's economic, social, and territorial cohesion. It aims to correct imbalances between countries and regions. It delivers on the Union's political priorities, especially those pertaining to green and digital transitions (European Commission 2023). The EU's Cohesion Policy is implemented through interventions.

There are three trends in the implementation of cohesion policies: technology, geographical integration, and institutional differences in the EU. Institutions are the key force according to the scholars analyzing this trend, because institutions shape the ability of an economy to use and develop its resources (Farole et al. 2011).

## 3. Hypotheses

The European Council on Foreign Relations (Puglierin and Zerka 2022) asserts that European sovereignty does not involve building fences or withdrawing from the world stage, nor should it be viewed as opposing national sovereignty. Rather, it involves en-

hancing the EU's ability to manage the intricate interdependencies that are characteristic of today's world. The Council insists that Europe should not be bullied by others. The sovereignty index produced by the European Council on Foreign Relations indicates that the sovereignty scores of countries are relatively low, indicating that they have shared a significant portion of their sovereignty with the EU's governing bodies and NATO.

Shared sovereignty can lead to conflicts associated with multilevel governance and a prolonged crisis of party-based democracy (Bickerton et al. 2022). Donohoe (2013) hopes that once the Troika departs, a member country will regain its economic sovereignty.

The state, as a political power, is always a viable positive and coercive force when it possesses economic power, institutions, and instruments. Delegating economic power to external bodies implies self-disabling and reduces the state's and the country's people's responsibility. Furthermore, voluntarily relinquishing sovereignty implies that the state fears that it cannot survive alone and must join neighboring states. Weak nations believe that having a larger partner is always safer. Weak states seek to join supranational unions because their political forces are so weak that they lack the confidence in their own abilities and competence to lead the country and withstand global pressures. Indeed, groups of weak countries within unions may survive together better than individual countries. However, non-member European nations did not face the same survival challenges as EU members since the EU's inception. Furthermore, member nations of a union may develop more quickly if that union is established correctly.

There are two categories of countries that are interested in founding a union. The first is represented by strong countries that want to involve many weaker countries into unions to make them an isolated marketplace for their goods. The second category is represented by weak nations that would be comfortable outsourcing their leadership responsibilities to the external unelected bodies, such as unions, and become passive actors. These weak political actors are unable to design the country's future or plan anything that could rebuild the country into another powerful nation. Additionally, the state is not sure that it will be able to protect itself from outside enemies and seeks to join a military alliance. The state ignores the principle of inseparability of economic and political power (Mayer and Phillips 2017).

The EU is the best example, the member nations of which delegated those inseparable powers to the EU and unsuccessfully tried to be politically independent of the Union. The establishment of the European Parliament cut member nations' political and economic sovereignty significantly and weakened member nations' political independence.

The European Parliament centralized much of the legislative functions of national governments. Heinz-Jürgen Axt (2011) stated that many scholars and policy makers argue that the EU is characterized by a democratic deficit. Andreas Føllesdal and Simon Hix (2005) have referred to the "standard version" of the democratic deficit. Heinz-Jürgen Axt (2011) writes about Simon Hix's points the following.

> It is assumed that executive powers are strengthened and legislative powers are weakened. The institutional setting of the EU has increased the influence of the European Commission and the Council whereas the national parliaments have lost the power to control the executive institutions. National parliaments are no longer in a position to monitor and control decision making in the EU effectively. Ministers who are part of the national executive power structures dominate decision making at EU level. And the European Council is not subject to democratic control. Simon Hix has presented this argument in the following statement: "Governments can ignore their parliaments when making decisions in Brussels or can be out-voted in the Council."

**Hypothesis 1.** *Shared sovereignty will be negatively associated with the member country's economic development.*

**Hypothesis 2.** Shared *sovereignty will moderate the relationship between member country political institutions and economic development.*

Economic sovereignty is hampered when a country lacks the power to make independent financial and trade decisions, particularly when it does not have its own currency. The sovereignty and conflict literature focuses on the more political aspects of this subject (Brack et al. 2021) through the theories of federalism, non-functionalism, intergovernmentalism, and post-functionalism. The economic root of political integration in Europe is to be investigated more from the point of cost-benefit analysis for every member nation. Economic integration cannot be achieved without political decisions. The political decision is to economically integrate into a union. When this integration happens, policy makers' power in member nations drastically diminishes and they start thinking and speaking about restricted economic and political sovereignty, and researchers discuss shared or pooled sovereignty (Keohane 2002). The single foreign trade policy restricts member nations' opportunities to explore new markets and take advantage of effective and efficient trade relations and capital investment in all countries in which the member country has interests. The common market and union compose a very restricted community and policies are not made in the interests of all the member nations because the unelected executive body will make subjective decisions under the influence of the lead countries.

There are lead members in every union. They have stronger economic and veto power (may be informal) to restrict member nations from adopting economic programs that are good for the people of those nations. There is always blame for not thinking of all the members of the union and being selfish and assertive. In the end, the protesting country will not be able to achieve anything significant to develop its economy and improve the lives of its citizens at higher rates or avoid disruptions in this development. Member nations cannot freely attract foreign investment or invest their capital anywhere else, purchase foreign-made products or export their products anywhere else without considering the interests of the Union.

A single currency could help achieve the goal of creating a single country. However, that country does not exist yet. Therefore, not having their own currency, member countries of the EU cannot use all the possible features of local money that an independent country can. They cannot do emission, which sometimes helps mitigate crisis situations. Debt financing programs to handle crises bury the country in external debt (the EU and its Central Bank are still external entities for member nations); the conditions of repayment are very harsh.

No amount of borrowed money can help if the country does not make holistic decisions for itself. The country's credit rating decreases, investment inflows dry up, and foreign trade will be under the control of the EU. When a government's formal political power is separated from its economic power it has no power at all; it does not make economic decisions for its own country because the political power has no economic instruments, and the country has no economic institutions that the union has.

**Hypothesis 3.** *A member nation's shared sovereignty will negatively moderate the relationship between foreign direct investment inflows and outflows and the nation's economic development.*

## 4. Methods

### 4.1. Data Sources and Measures

The present study utilizes quantitative data on member nations' macroeconomic parameters (GDP growth rates in 2015 and 2019, %; per capita GDP in 2015 and 2019, $) of all of the 27 member countries (the UK is excluded) of the EU from publicly available sources of The World Bank (2019a, 2019b), the Freedom House (The Political Freedom Index (2019): see the weblink with this name in the references), (The Fraser Institute (2019): see the weblink with this name in the references), (The Heritage Foundation (2019): see the weblink with this name in the references), the European Council of Foreign Relations (European Sovereignty Index: see the weblink with this name in the references), and various academic and internet sources.

The political freedom index is a survey of public opinion that the Freedom House collects annually. The survey is conducted on a 100-point scale, 100 being free and 0 is not free. This scale is divided into two subscales: political rights (possible 60 points) and civil liberties (possible 40 points). The economic freedom index in both institutions (the Fraser Institute and the Heritage Foundation) is generated by surveying the public in all involved countries on a 100-point scale, with 100 points being free and 0 being not free. The subindices have the same 100-point scale. The European Sovereignty Index (2019) uses primary sources (input from our 27 national associate researchers) and secondary sources (such as public opinion data, official statistics, and other rankings). The scale is 0–10, 10 being an excellent sovereignty score and 0 being no sovereignty. The state of the sovereignty data belongs to a fixed time, the beginning of 2019. All economic data belong to the end of 2019. The institutional data belong to the beginning of 2019. All data are numerical and compatible with statistical analyses. All data are one-directional and positive.

Economic development: GDP growth rates measure economic development in the EU member nations.

Political institutions: the political freedom aggregate index, political rights, and civil liberties measure the political institutions of the member nations of the EU. The economic development data was chosen for the most stable period, 2015–2019, unaffected by recessions and turbulence in the world economy.

Economic institutions: the economic freedom index and its subindices in the Fraser Institutions and the Heritage Foundation's surveys measure the economic institutions of a member nation.

Sovereignty: the European Sovereignty Index measures the sovereignty of the EU member nations.

Foreign economic relations: inbound and outbound foreign direct investment and foreign trade (imports and exports) measure foreign economic relations. The FDI is measured by its share in the GDP as well as in US dollars. Imports and exports are measured in US$.

### 4.2. Dependent, Independent, and Moderating Variables

The dependent variable is GDP growth rates in a five-year range, 2015–2019, for the member nations of the EU. Independent variables are the sovereignty index, the political freedom index, and the economic freedom index, and their subindices, country cultural dimensions, foreign direct investment, and foreign trade (imports and exports). The moderating variable is the sovereignty index.

### 4.3. The Model and Data Processing Techniques

The purpose of my study is to reveal the causal relationships between independent and dependent variables using the instrumental variable estimation and direct and moderating (interaction) effects of independent variables in multiple regression. The model (Figure 1) involves the effects of political and economic institutions on economic development under the effects of sovereignty indices as instrumental variables. If sovereignty indices are weak instruments, then simple moderation can reveal the effects of the current conditions of shared sovereignty on institutions, foreign direct investments, and foreign trade. In the instrumental variable estimation (Angrist and Krueger 2001), sovereignty was not a strong instrument in determining causal relations between economic development and political and economic institutions and foreign economic relations (foreign direct investment and trade). Therefore, the direct impact of independent variables on the dependent variable and the moderating effects of sovereignty on the relationship between economic development and institutions and foreign economic relations will be examined in correlation analyses and OLS (Stigler 1981) estimations.

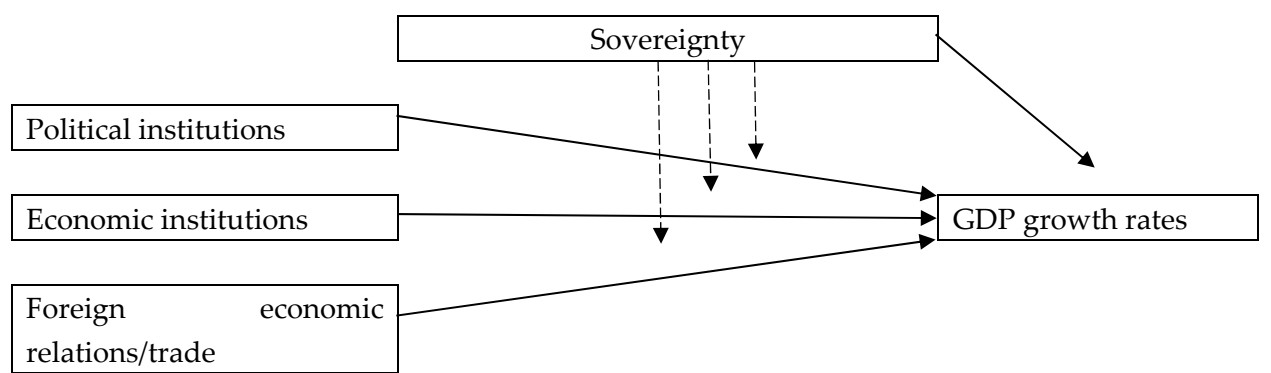

**Figure 1.** Direct and moderating effects of country factors on economic development. ⟶
Direct effects – – – – – ▶ Moderating effects.

I used the following multiple regression equation (Fisher 1938) for hypothesis testing.

$$Y = b_0 + b_1X_1 + b_2X_2 + \ldots + b_nX_n + c.$$

I utilized the following model for testing moderating effects (Kenny and Judd 1984) of
the third variable.

$$Y = i + aX + bM + cXM + e.$$

Chi-square goodness of fit test for data independence. All data are categorical quanti-
tative ordinal or numerical.

## 5. Analysis and Results

Analyses involve member states' sovereignty, institutions, intranational economic rela-
tions, and economic development relationships within the EU. The European sovereignty
index has several subindices, namely, economic, defense, climate, health, technological,
and migration sovereignties. Sovereignty correlates with several country variables. Most
importantly, sovereignty correlates negatively and significantly with key economic de-
velopment indicators, namely with annual GDP growth rates, annual GDP per capita
growth rates, five-year GDP growth rates, five-year GDP per capita growth rates, the size
of government, government spending, transfers and subsidies, and total marginal tax/tax
burden. The negative relationships among key variables indicate that shared sovereignty
does not facilitate desired economic development in member nations (Table 1).

**Table 1.** Correlations among the sovereignty index, institutional, economic, and cultural factors in
member countries.

| | sog [1] | gsp | ts | tmt | pd | gdpgr | gdppcgr | gdpgr5 | gdppcgr5 |
|---|---|---|---|---|---|---|---|---|---|
| National sovereignty score | −0.62 ** | −0.61 ** | −0.62 ** | −0.63 ** | −0.46 * | −0.62 ** | −0.75 ** | −0.53 ** | −0.63 ** |
| | taxb | govs | pr | cl | lspr | ji | ppr | rtb | br |
| National sovereignty score | −0.78 ** | −0.50 ** | 0.59 ** | 0.58 ** | 0.66 ** | 0.75 ** | 0.59 ** | 0.40 * | 0.69 ** |
| | indul | gdppc | imp | exp | tb | gdp | govi | bf | inf |
| National sovereignty score | 0.61 ** | 0.59 ** | 0.63 ** | 0.63 ** | 0.50 ** | 0.59 | 0.80 ** | 0.74 ** | 0.38 * |
| | finf | cpi | je | propr | sb | ind | | | |
| National sovereignty score | 0.41 * | 0.82 ** | 0.78 ** | 0.73 ** | 0.49 ** | 0.42 ** | | | |

[1] Please see the description of abbreviations in Tables 1–3 below. ** Correlation is significant at the 0.01 level
(2-tailed). * Correlation is significant at the 0.05 level (2-tailed).

**Table 2.** Correlations among the economic sovereignty index, institutional, economic, and cultural factors in member nations.

|  | sog | gc | tmt | pd | ua | gdpgr | fdii | taxb | fdiin |
|---|---|---|---|---|---|---|---|---|---|
| Economic sovereignty score | −0.47 * | −0.68 ** | −0.46 * | −0.50 ** | −0.48 ** | −0.38 * | −0.43 * | −0.64 ** | −0.40 * |
|  | gdppc5 | demi | pr | cl | gi | lspr | ji | ppr | reg |
| Economic sovereignty score | −0.48 * | 0.40 * | 0.46 * | 0.46 * | 0.43 * | 0.70 ** | 0.73 ** | 0.67 ** | 0.44 * |
|  | br | ind | indul | gdppc | exp | imp | tb | gdp | ecf |
| Economic sovereignty score | 0.70 ** | 0.56 ** | 0.67 ** | 0.70 ** | 0.55 ** | 0.55 ** | 0.44 * | 0.45 * | 0.55 ** |
|  | propr | je | govi | bf | inf | cpi |  |  |  |
| Economic sovereignty score | 0.78 ** | 0.74 ** | 0.82 ** | 0.67 ** | 0.54 ** | 0.82 ** |  |  |  |

** Correlation is significant at the 0.01 level (2-tailed). * Correlation is significant at the 0.05 level (2-tailed).

**Table 3.** Correlations among the economic development indicators, shared sovereignty, institutional, economic, and cultural factors in member countries.

|  | sov | sovc | sovd | sovh | pr | sg | ts | tmt | ppr |
|---|---|---|---|---|---|---|---|---|---|
| GDP growth rates in 2019 compared to 2015 | −0.53 ** | −0.40 * | −0.62 | −0.51 ** | −0.40 * | 0.65 ** | 0.85 ** | 0.57 ** | 0.41 * |
|  | cc | gdpgra | gdppcgr | fdii | exp | gdp | taxb | govs | gdppc5 |
| GDP growth rates in 2019 compared to 2015 | 0.41 * | 0.89 ** | 0.61 ** | 0.43 * | −0.39 * | −0.45 * | 0.59 ** | 0.68 ** | 0.67 |
|  | sov | sovc | sove | sovh | sovt | pr | cl | sog | gc |
| Per capita GDP growth rates in 2019 compared to 2015 | −0.63 ** | −0.66 ** | −0.48 * | 0.69 ** | −0.49 ** | −0.43 * | −0.47 * | 0.75 ** | 0.48 * |
|  | ts | tmt | lspr | ji | ppr | cc | lmr | pd | indul |
| Per capita GDP growth rates in 2019 compared to 2015 | 0.68 ** | 0.70 ** | −0.41 * | −0.47 * | −0.40 | 0.47 * | 0.44 * | 0.37 * | −0.71 |
|  | gdpgr | gdppc | gdp5 | propr | je | govi | taxb | govs | bf |
| Per capita GDP growth rates in 2019 compared to 2015 | 0.68 ** | −0.53 ** | 0.67 ** | −0.41 * | −0.46 | −0.53 | 0.85 ** | 0.70 ** | −0.42 |
|  | cpi |  |  |  |  |  |  |  |  |
| Per capita GDP growth rates in 2019 compared to 2015 | −0.56 ** |  |  |  |  |  |  |  |  |

* $p < 0.05$; ** $p < 0.001$.

Description of abbreviations used in Tables 1–3.

| sog | size of government |
|---|---|
| gsp | government spending (Fraser Institution) |
| ts | transfers and subsidies |
| tmt | total marginal tax |
| pd | power distance |
| gdpgr | GDP growth rates (2019/2018) |
| gdppcgr | GDP per capita growth rate (2019/2018) |
| gdpgr5 | GDP growth rates (2019/2015) |
| gdppcgr5 | GDP per capita growth rates (2019/2015) |
| taxb | tax burden |
| govs | government spending (Heritage Foundation) |
| pr | political rights |
| cl | civil liberties |
| lspr | legal system and property rights |
| ji | judicial integrity |
| ppr | protection of property rights |
| rtb | regulatory trade barriers |
| br | business regulation |
| indul | indulgence |
| gdppc | GDP per capita |
| imp | imports |
| exp | exports |
| tb | trade balance |
| gdp | GDP current US$ |
| govi | government integrity (Fraser Institution) |
| bf | business freedom |
| inf | investment freedom |
| finf | financial freedom |
| cpi | corruption perceptions index |
| je | judicial effectiveness |
| propr | property rights (Fraser Institution) |
| sb | ease of starting business |
| ind | individualism |
| gc | government consumption |
| ua | uncertainty avoidance |
| fdii | FDI inflows, % in GDP |
| fdiin | FDI inflows, current US$ |
| demi | democracy index |
| gi | government integrity (Heritage Foundation) |
| reg | regulations |
| ecf | economic freedom (Heritage Foundation) |
| sov | sovereignty |
| sovc | climate sovereignty |
| sovh | health sovereignty |
| sovt | technological sovereignty |
| cc | capital controls |
| lmr | labor market regulations |

Economic sovereignty had significant negative correlations with the size of government, government consumption, top marginal tax, power distance, uncertainty avoidance, annual GDP growth rates, FDI inflows (% in GDP), FDI inflows (current US$), and GDP per capita in 2019 compared to 2015. Economic sovereignty did not have any correlation with the five-year GDP growth rates (Table 2). The EU's social/entitlement programs and supported lifestyles contribute to the EU citizens' indulgence.

Not only does overall shared sovereignty have negative correlations with key member state variables, but also its subcategories have such correlations with the GDP growth rates in 2019 against 2015 and other country variables (Table 3) such as GDP per capita growth rates in 2019 against 2015. This evidence indicates that state sovereignty is shared excessively with the EU so that it has negative relations with economic development. Only

the sovereignty subindex migration had no significant correlation with either 2019 GDP growth rates against 2015 or 2019 GDP growth rates per capita against 2015.

The analysis for moderation indicates that *shared sovereignty* moderates the relationship between *economic development* and member country *political institutions*. The interaction effects of political rights and sovereignty were statistically significant at $p < 0.05$ and negative (Table 4), indicating that the shared sovereignty's negative impact on economic development becomes stronger with diminishing political rights. As seen in Tables 5 and 6, shared sovereignty has a significant negative impact on the economic development of member states. The effect size measured by semi-partial correlations is medium (r = 0.30 indicates a medium effect).

**Table 4.** Moderating effects of shared national sovereignty on the relationship between economic development and political institutions (Dependent variable: GDP growth rates in 2019 against 2015).

| Independent Variables | B | Standard Error | Beta | t | Sig. | Zero-Order | Correlations | |
|---|---|---|---|---|---|---|---|---|
| | | | | | | | Partial | Semipartial |
| (Constant) | −254.93 | 137.18 | | −1.86 | 0.076 | | | |
| Political rights | 40.94 | 19.71 | 3.29 | 2.08 | 0.049 | −0.529 | 0.388 | 0.332 |
| Shared sovereignty | 67.90 | 32.63 | 9.23 | 2.08 | 0.049 | −0.403 | 0.397 | 0.322 |
| Political rights*Sovereignty | −10.20 | 4.67 | −12.02 | −2.18 | 0.039 | −0.551 | −0.414 | −0.349 |
| F = 5.42 *; $R^2$ = 0.644 | | | | | | | | |

* $p \leq 0.05$.

**Table 5.** Direct effects of political institutions and shared sovereignty on economic development.

| Independent Variables | B | Standard Error | Beta | t | Sig. | F | Sig. | $R^2$ |
|---|---|---|---|---|---|---|---|---|
| Political rights | −1.77 | 2.64 | −0.142 | −0.067 | 0.511 | 0.497 | 0.016 | 0.293 |
| Shared sovereignty | −3.28 | 1.56 | −0.456 | −0.21 | 0.046 | | | |

Dependent variable: GDP growth rates in 2019 against 2015.

**Table 6.** Direct effects of sovereignty on economic development.

| Independent Variables | B | Standard Error | Beta | t | Sig. | F | Sig. | $R^2$ |
|---|---|---|---|---|---|---|---|---|
| (Constant) | 34.45 | 6.81 | | 5.06 | 0.001 | 9.70 | 0.005 | 0.280 |
| Shared sovereignty | −3.89 | 1.25 | −0.529 | −3.12 | 0.005 | | | |

Dependent variable: GDP growth rates in 2019 compared to 2015.

Direct effects of other shared sovereignty categories on economic development suggest that key categories (climate change, national defense, and healthcare) are significant and negative (Table 7). Membership in NATO takes away the majority of sovereignty in national defense. Moderating effects of sovereignty also were discovered in the relationship between FDI inflows and economic development (Table 8). The positive B-coefficient for the interaction predictor indicates that economic development will be weaker if sovereignty decreases and FDI outflows from member states increase. The effect sizes of the predictors range from medium to large (r = 0.30 or higher but lower than 0.50 indicates medium effect: r = 0.481 is close to the large effect).

**Table 7.** Direct effects of specific sovereignty indices on economic development (Dependent variable: GDP growth rates in 2019 compared to 2015) (N = 27).

| Independent Variables | B | Standard Error | Beta | t | Sig. | F | Sig. | $R^2$ |
|---|---|---|---|---|---|---|---|---|
| (Constant) | 25.16 | 8.26 | | 3.05 | 0.006 | | | |
| Shared national sovereignty in: | | | | | | 7.53 | 0.001 | 0.693 |
| climate change | −3.89 | 1.35 | −0.62 | −2.89 | 0.009 | | | |
| national defense | −1.93 | 0.67 | −0.47 | −2.90 | 0.009 | | | |
| the economy | 2.19 | 1.18 | 0.044 | 1.85 | 0.079 | | | |
| healthcare | −2.85 | 1.39 | −0.58 | −2.05 | 0.053 | | | |
| migration | 2.36 | 1.73 | 0.21 | 1.37 | 0.187 | | | |
| technology | 2.10 | 1.23 | 0.41 | 1.71 | 0.103 | | | |

**Table 8.** Moderating effects of shared national sovereignty on the relationship between economic development and FDI outflows (Dependent variable: GDP growth rates in 2019 against 2015).

| Independent Variables | B | Standard Error | Beta | t | Sig. | Zero-Order | Correlations | |
|---|---|---|---|---|---|---|---|---|
| | | | | | | | Partial | Semipartial |
| (Constant) | 25.84 | 6.49 | | 3.98 | 0.001 | | | |
| Sovereignty index | −2.52 | 1.16 | −0.34 | −2.16 | 0.041 | −4.92 | −0.411 | −0.314 |
| FDI outflows, % in GDP | −0.17 | 0.06 | −0.99 | −2.82 | 0.010 | 0.170 | −0.506 | −0.409 |
| Sovereignty index * FDI outflows | 0.05 | 0.02 | 1.18 | 3.31 | 0.019 | 0.42 | 0.568 | 0.481 |

F = 8.10 at $p$ = 0.001; $R^2$ = 0.514.

Direct effects of FDI inflows on economic development are positive and significant, while FDI outflows have a significant negative impact on the economic development of member states (Table 9).

**Table 9.** Direct effects of FDI inflows and outflows (% in GDP) on economic development (Dependent variable: GDP growth rates in 2019 compared to 2015).

| Independent Variables | B | Standard Error | Beta | t | Sig. | F | Sig. | $R^2$ |
|---|---|---|---|---|---|---|---|---|
| (Constant) | 11.83 | 1.02 | | 11.66 | 0.001 | | | |
| FDI outflows | −0.21 | 0.061 | −1.21 | −3.40 | 0.002 | 9.83 | 0.001 | 0.450 |
| FDI inflows | 0.29 | 0.067 | 1.52 | 4.29 | 0.001 | | | |

## 6. Discussion

### 6.1. Findings

Hypothesis 1, stating that shared sovereignty will be negatively associated with the member country's economic development is supported because the aggregate sovereignty index and its main subindices (climate, defense, and health) are significantly and negatively correlated and associated (in the OLS estimation) with economic development of member nations. Shared sovereignty negatively moderates the relationships between the political institutions and economic development, supporting Hypothesis 2. Hypothesis 3, suggesting that a member nation's shared sovereignty will moderate the relationship between foreign direct investment inflows and outflows and the nation's economic development, is partially supported regarding FDI outflows. FDI outflows negatively impact economic growth, but shared sovereignty in interaction with negatively impacting FDI outflows worsens this impact in member nations.

### 6.2. Limitations of the Study and Future Research

The present study did not analyze the specific details of the political systems of member countries and the EU, which established its supremacy over the member nations. Additionally, there is no focus on the significant positive achievements of the EU in this study. Therefore, a more holistic analysis of the viability of the EU as an economic and political system involving more qualitative and quantitative data will help. Research should continue analyzing the status of democracy and the economic system in the member nations (Bickerton et al. 2022).

EU officials insert themselves into the governance of member nations. This lack of democratic accountability has been causing serious problems for member states and the system of governance in the EU (Hix and Hoyland 2022). Governments that resist pressure from the EU are called activist governments (Gabrisch and Werner 1998). This issue is also important for future research.

The viability of the single currency and the policies of the Central Bank of the EU (Fairless 2013) should be re-examined. Having one's own currency gives a country a wide latitude to program its national economy. The EU took away such an opportunity from the Eurozone countries.

Future research should analyze three scenarios for the EU's future. The first, the EU stays as it is. The second, the EU turns into a trade agreement only in the form of the European Trade Council as Winston Churchill (1946) suggested. The third, the EU is considered as a single country, a federation.

Even though the data came from member countries, the results were the outcome of the entire sample; EU countries served as observations in the sample. Therefore, specific results by country were not identified. Future research should pay attention to particular countries and make specific recommendations.

### 6.3. Research and Practical Implications

The results of the present study will be beneficial for member nations of the EU and research institutions to further investigate the status of national and shared sovereignty and economic growth in the EU. The EU has problems related to its underserved peripheral countries. The most critical issue is the legitimacy of the EU, the governing bodies of which tightly govern the member nations without any constitutional authority (Isiksel 2016). Furthermore, the EU isolates member nations from the external world and monopolizes the European market (Von Der Burchard et al. 2019; Piekutowska and Marcinkiewicz 2020). Governments in member nations should address the issues that impede the freedom of trade and investment. They also should revise their sovereignty and institutional setup to optimize them and facilitate effective governance, reasonable regulation, effective trade policy, financial stability, and rapid economic growth.

### 7. Conclusions

The European Union is an impressive political, economic, and social entity, but its existence has brought about a significant reduction in member nations' state sovereignty and independence. Consequently, true political competition among parties within member nations is often lacking. The lack of competing ideas that can be independently implemented without the EU's involvement has resulted in sluggish economic development. The absence of real competition among political factions is especially evident in the Eurozone, where political forces are unable to execute significant projects without the EU's approval, thereby hampering member nations' independence. Eurozone member states are more vulnerable to losing their sovereignty due to the over-delegation of political and economic power to the EU. This over-delegation often results in the inability of member nation governments to make necessary political, economic, and social decisions on domestic affairs and foreign relations. As a result, a member nation ceases to be a fully sovereign entity and becomes more of a semi-province of a larger territorial system known as the Union, which is not yet a federation (Hix and Hoyland 2022).

While Euroscepticism is reflected in emotion-laden public discourse (Fanoulis and Guerra 2017), complex data show that the founding of the EU aimed to achieve socio-economic goals for building strongly socially oriented societies that are inconsistent with the capitalist incentive system. This has led to a transformation that results in lowered productivity, effectiveness, and efficiency, as people are not motivated to work as they are in a capitalist system.

The present study's quantitative analyses reveal that the sovereignty index and its key sub-indices, including defense sovereignty, healthcare sovereignty, and climate sovereignty, negatively correlate with and impact economic growth. Additionally, the sovereignty index moderates the significant negative relationships between economic development, political institutions (specifically political rights), and FDI outflows. As a result, shared sovereignty has negative direct and moderating effects on economic growth. The EU should consider problems of sovereignty of institutional, and of economic development revealed in the present research in their further integration processes through the Cohesion Policies (as discussed in Farole et al. 2011).

National political parties should clarify to their citizens the truth about this conglomerate (Fanoulis and Guerra 2017), which restricts political rights and holds countries back from rapid economic development. The current setup of the EU needs to be revised, and member nations must find the correct solutions to their institutional problems to protect true democracy and achieve rapid economic development. This solution may involve weakening the legislative and executive functions of the European Union or establishing a true federation where countries lose their independence and become provinces of the federation. Consequently, the political system will change.

**Funding:** This research received no external funding.

**Informed Consent Statement:** Not applicable.

**Data Availability Statement:** Data are publicly available in the sources listed in the references.

**Conflicts of Interest:** The author declares no conflict of interest.

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
