# Peer review of "Issues of EU Member Nations’ Shared Sovereignty, Institutions, and Economic Development"

_economies, doi:10.3390/economies11040128_

Round 1

Author Response

Thank you very much for reviewing my manuscript and for your feedback.

  1. An English language professor proofread the manuscript.
  2. I improved the Introduction. The addition is in red.
  3. I expanded the methods section. The research design is standard. 
  4. Thank you for your "yes" on the methods.
  5. Thank you for your "yes" to the research results.
  6. I improved the conclusions section.
  7. My creation in doing this work concerning the qualitative literature review, methodology, and findings are to reduce the literature gap that rarely investigates the quality of EU institutions. Most researchers accept the EU as it is. The EU had significant problems. One is very slow economic growth and its reasons; I have explained this concept throughout the paper. The findings are that shared/reduced national sovereignty has a significant negative direct and moderating impact on critical institutional and economic development variables.
  8. I cited additional literature, including recent ones, and included them in the list of references.
  9. I also improved the implications for the research and practice section.
  10. Thank you for the positive assessment of the results.

Please find the revised manuscript in the attachment. The added parts of the paper are in red. Added recent references are in blue.

Reviewer 2 Report

Dear authors,

I have made my revision. Please consider cearfully my comments and address with high consideration

Author Response

Thank you very much for reviewing my manuscript and for your valuable feedback.

My responses to your comments:

  1. An English language professor proofread the manuscript.
  2. I added several relevant references, including recent ones. The recent ones are in the red area of the text. Also, I typed recent references in blue.
  3. I added more references relevant to the research.
  4. The research design is standard.
  5. I edited the Methods section to explain the data characteristics. Also, I have presented the model estimation techniques.
  6. The results are in the tables and narration. The tables and narration do not duplicate. They should be all right.
  7. I changed and extended the Conclusions section.
  8. Rationale: 

      a.  The focus was the EU with country data utilization. Every country was used as one observation. Therefore, there are no country-specific outcomes from this study. The country data shows the conditions of the EU’s economic development overall. Yes, the paper has this limitation, and I indicated this limitation in my revision.

      b. Almost all vital institutional factors were utilized. You can see them in the footnotes of the tables. I also explained the data characteristics and sources in the Methods section.

      c. I agree with you. I have indicated this statement in my revision.

      d. Overall: I agree with you. I added a “institutions and economic development” section and used your recommended sources, among others.

9. Literature review: I utilized all your recommended literature in my revision and added several recent sources.

10. I extended the theoretical framework by adding a section, “Institutions and economic development.”

11. Methodology: thank you for accepting this section.

12. Empirical findings: thank you for your agreement with the findings.

13. Conclusion: I have extended it.

14. Referencing: I added some more per your comments.

Please find the revised manuscript in the attachment. The added parts of the paper are in red. Added recent references are in blue.

Reviewer 3 Report

The issue is very relevant, could be cognitively interesting and useful in practice. However, the research are not fully realised. The results cannot be considered objective.

1. The article contains many unambiguous statements (these) that are not scientifically proven. These are probably the beliefs of the author/authors. These include, e.g.:

- The EU is the best example, the member nations of which delegated those inseparable powers to the EU and unsuccessfully tried to be politically independent of the Union. However, they always remain politically dependent: whatever the union orders, the member country must do in political, economic, and social fields (181-185)

- EU officials insert themselves into the governance of member nations: this lack of democratic accountability has been causing serious problems for member states and the system of governance in the EU (383-385);

- Also, the EU isolates member nations from the external world and monopolizes the European market. Governments in member nations should address the issues that impede freedom of trade and investment (402-404)

2. The title is inadequate to the content of the article. The article refers to the limitation of the sovereignty of EU member states that prevents their economic growth and development.  So what institutional issues are at stake?

3. There are many uncertainties regarding the empirical studies:

- what specific data were included? are the indices and indicators developed by different institutions on the basis of different assumptions and methods? are the data comparable?

- for which countries was the research carried out? all in the EU or a selection?

- Why only for the period 2015-2019? - If a loss of sovereignty is assumed to occur, it occurs in multiple stages over a longer period of time. The period adopted is far too short.

A precise explanation of the sources and tools of the study is needed.

4. The argumentation carried out is one-sided, although in some parts it has references to the literature on the subject. However, only to those items that support the assumptions of the author(s). There is no indication that there are other opinions or research evidence. There are, after all, works showing that EU membership accelerates economic growth and development in member states. The authors are silent about this. It should be shown that such studies are known to the author(s) and counter-arguments should be shown.

I would suggest directing the article to a journal with a different focus, such as a political science journal. Line 73 states:  'The present study concurs with the classical theory of state sovereignty, which is based on Westphalian or (Vattelian) sovereignty. This indicates a non-economic context for the study. Furthermore, most of the literature cited is about politics, law and not economics.

Author Response

Thank you very much for reviewing my manuscript and for the valuable feedback.

All the changes and additions are in red within the manuscript. Recent references are in blue. Please see the attachment.

My responses to your comments:

  1. An English language professor proofread the manuscript.
  2. I added several relevant references, including recent ones. The recent ones are in the red area of the text. Also, I typed recent references in blue.
  3. I added more references relevant to the research.
  4. The research design is standard.
  5. I edited the Methods section to explain the data characteristics. Also, I have presented the model estimation techniques.
  6. The results are in the tables and narration. The tables and narration do not duplicate. They should be all right.
  7. I changed and extended the Conclusions section.
  8. Ambiguous statements (181-185; 383-385; 402-404): I cited sources to support them and removed one.
  9. I changed the title of the paper. The institutional issues are political rights, civil liberties, and economic freedom measured by the economic freedom index and its subindices in the correlation and other tables.
  10. All 27 EU member countries’ data were utilized. The UK was not among them.
  11. Data are comparable: numerical and one-directional (positive). I additionally described data characteristics in the methods section.
  12. The years are not 2015-2019 but 2019 against 2015. A single point of time is used (2019). The sovereignty index and economic and political freedom indices were by the beginning of 2019, and economic data (GDP and FDI) were by the end of 2019. I explained this fact in the paper, adding a statement.
  13. The focus is on the problems of the EU rather than its achievements. A lot of literature has investigated the pros of the EU. Literature has a significant gap between pros and cons. I added a literature review on the positives of the EU in the Theoretical Framework section. Literature indicates that membership does accelerate economic development, but not significantly. That is a problem. I added citations in the text.
  14. The paper is about economic development under the impact of shared sovereignty and country institutions. Therefore, I submitted the paper to this journal. I hope you will approve it.

Author Response

Thank you very much for reviewing my manuscript.

All the changes and additions are in red within the manuscript. Recent references are in blue. Please see the attachment.

My responses to your comments:

  1. An English language professor proofread the manuscript.
  2. I added several relevant references, including recent ones. The recent ones are in the red area of the text. Also, I typed recent references in blue.
  3. I added more references relevant to the research.
  4. The research design is standard.
  5. I edited the Methods section to explain the data characteristics. Also, I have presented the model estimation techniques.
  6. The results are in the tables and narration. The tables and narration do not duplicate. They should be all right.
  7. I changed and extended the Conclusions section. The results support conclusions.
  8. I removed passages that were controversial and subjective.
  9. I expanded the introduction, added recent references, and added a subsection to the Theoretical Framework section.
  10. I revised the manuscript thoroughly based on other reviewers’ comments.
  11. I explained data sources and characteristics better.
  12. I made all the cited references relevant to the research.
  13. I have changed the title of the manuscript.
  14. I indicated the focus of the project: issues and problems of the EU rather than its achievements.

Round 2

Reviewer 1 Report

Since the authors have made changes based on my suggestions, I think this article should be published.

Reviewer 2 Report

Dear Author/Authors,

I have read carefully your review article. For me this form of mauscript is ok. 

Thank you very much for your good job. 

Reviewer 3 Report

I have read the new manuscript and explanations. I appreciate the changes made. They improved the understanding of research objectives and the research process. In this form, the article can be published.

Reviewer 4 Report

In view of the changes made by the author, which greatly improve its quality, I am of the opinion that the paper can be published in its present form.